# Diagnosis and management of jejunoileal diverticular haemorrhage: An update on the experience in a single centre

Hsuan-An Su[1]☯, Yu-Chun Hsu[2]☯, Fu-Yuan Siao[3], Hsu-Heng Yen [ID][2,4,5,6]*

**1** Department of Medical Education, Kaohsiung Chang Gung Memorial Hospital, Kaohsiung, Taiwan, **2** Endoscopy Center, Changhua Christian Hospital, Changhua City, Taiwan, **3** Department of Emergency Medicine, Changhua Christian Hospital, Changhua City, Taiwan, **4** General Education Center, Chienkuo Technology University, Changhua City, Taiwan, **5** College of Medicine, Chung Shan Medical University, Taichung City, Taiwan, **6** Taiwan Association for the Study of Small Intestinal Diseases (TASSID), Taipei, Taiwan

☯ These authors contributed equally to this work.
* blaneyen@gmail.com

## Abstract

### Introduction

Jejunoileal diverticular haemorrhage is a rare disease that is difficult to diagnose and treat. Despite advances in endoscopic technology, recommendations on diagnosis and management for jejunoileal diverticular haemorrhage have remained unchanged and these new options have not been compared against traditional surgical management.

### Materials and methods

We retrospectively reviewed the diagnosis, management, and outcome for jejunoileal diverticular haemorrhage cases at our institution over the past 20 years. Data were organized and analysed by chi-square test, student *t*-test and Kaplan–Meier survival analysis.

### Results

The most utilised diagnostic procedure was computed tomography, followed by enteroscopy, angiography, small bowel flow-through and surgery. Primary treatments included, in a decreasing order, medical therapy, surgery, endoscopy and radiology. Surgical treatment was not associated with rebleeding, but it did result in longer hospital stays and larger blood transfusions than non-surgical treatments. The bleeding-related mortality rate was very low. Notably, there was also little change in the diagnosis and treatment between decades.

### Conclusion

We presented our experience with the diagnosis and management of jejunoileal diverticular haemorrhage, as well as long-term follow-up after treatments that have not been reported previously. Surgical treatment continues to dominate management for jejunoileal diverticular haemorrhage, but we support increasing the role of endoscopy for select patient groups.

**Data Availability Statement:** All relevant data are within the paper and its Supporting Information files.

**Funding:** The authors received funding from Changhua Christian Hospital (http://www2.cch.org.tw/cch_english/) for this work: numbers 106-CCHIRP-030 and 108-CCHIRP-018 for Dr Hsu-Heng Yen, and number 105-CCH-IRP-071 for Dr Fu-Yuan Siao. The funder played NO role in the study design, data collection and analysis, decision to publish, or preparation of the manuscript.

**Competing interests:** No authors have competing interests.

## Introduction

Diverticulosis of the small intestines, other than Meckel's diverticulum, is very rare. Its prevalence has been estimated at just 0.01%–2.3% in clinical studies, but rising to 4.6% in an autopsy study [1]. The most common site of small intestinal diverticulosis is the duodenum, followed by the jejunum and ileum [2]. Although haemorrhage is the most challenging and potentially fatal complication, it can be difficult to diagnose, especially when affecting the jejunum and ileum [1, 3, 4]. Obscure gastrointestinal bleeding due to diverticular disease occurs more often in Eastern (6.8%) than in Western (1.2%) populations [5].

Enteroscopy is the current primary endoscopic diagnostic and therapeutic approach for obscure gastrointestinal bleeding [6] including jejunoileal diverticular haemorrhage (JIDH). Alternative non-endoscopic diagnostic tools, including small bowel follow-through studies, computed tomography (CT), angiography and technetium red cell-tagged scans, are used when enteroscopy is not available [1]. Even resorting to exploratory laparotomy may fail to identify the site of bleeding [1]. Currently, surgical treatment after radiological examination remains the standard of care for JIDH [4, 7–10]. However, with the advent and popularisation of device-assisted endoscopic techniques (e.g. double-balloon enteroscopy), both endoscopic diagnosis and treatment have become possible [4, 11–15]. To date, comparisons of the different outcomes from surgical and non-surgical modalities are lacking.

In the present retrospective study, which follows on from earlier research [4], we reviewed the clinical features, diagnostic methods, treatments and outcomes of JIDH at our institute over the past two decades. We also report on current practice at our hospital and review the surgical and non-surgical management of this rare disease.

## Materials and methods

### Study design

We conducted a retrospective review of the medical records of patients with small intestinal diverticular haemorrhage who were diagnosed and treated at Changhua Christian Hospital, Changhua, Taiwan. The study covered two decades from 1st January 2000 to 31st July 2019. The study protocol was approved by the Institutional Review Board of Changhua Christian Hospital (No. 190814), and documentation of informed consent was waived because the study was conducted retrospectively.

### Participants and data collection

We included patients diagnosed with small intestinal diverticular haemorrhage (ICD-9 562.03 or ICD-10 K57.11). Any patient with bleeding from a duodenal diverticulum or a Meckel's diverticulum was excluded. Some of the patients in the present data set have been included in an earlier report [4]. For the purpose of our analysis, JIDH was diagnosed by the following criteria: (1) the presence of active bleeding at surgery or endoscopy; (2) active contrast extravasation on angiography or CT scan or (3) evidence of stigmata of recent haemorrhage at surgery, endoscopy or radiological examination in the absence of bleeding from other gastrointestinal sites.

Data from the endoscopic database and medical records at our hospital were reviewed and extracted. The following characteristics of patients with JIDH were extracted from the database, including sex, age, underlying disease (e.g. hypertension, diabetes mellitus, chronic renal failure, ischaemic heart disease and cerebrovascular disease), oral medication use (e.g. non-

steroidal anti-inflammatory or antiplatelet drugs) haemoglobin concentration on admission, length of hospitalisation, blood transfusion requirements (total units), time from symptoms onset to diagnosis, follow-up duration from diagnosis to the last visit, symptoms and signs, presence of hypovolemic shock on admission, rebleeding events and mortality. Included diagnostic methods were CT scan, endoscopy, angiography, small bowel flow-through and surgery. All of the CT scans in the present study were performed with CT angiography protocol. Included primary treatments were defined as surgical, radiological, endoscopic or supportive. The methods of enteroscopy include push endoscope (SIF-Q140, Olympus Co., Japan) performed in 5 cases before 2004, and double-balloon endoscope (EN-450P5 or EN-450T5, Fujinon Co., Japan) performed in the rest of the cases after 2004 in our institution.

## Outcomes

The primary outcome was the change in diagnostic or therapeutic management of JIDH at our institution over the last 20 years. Specifically, we compared the 2000–2009 period and the 2010–2019 period. The secondary outcomes were to compare the length of hospitalisation, the complication rate, the long-term rebleeding rate, the bleeding-related survival rate and the non-bleeding-related survival rate between treatment approaches.

## Statistical analysis

The acquired data were organised with Microsoft Excel and all statistical analyses were performed using MedCalc for Windows, version 18.11 (MedCalc Software, Ostend, Belgium; https://www.medcalc.org). Quantitative data are presented as means ± standard deviations. Statistical differences were assessed with the chi-square test for categorical variables or the student $t$-test for continuous variables. Kaplan–Meier survival curves were drawn to compare rebleeding rates between the surgically and non-surgically treated groups. Results were considered statistically significant for $p$-values of $<0.05$.

## Results

### Clinical features and characteristics of jejunoileal diverticular haemorrhage

The clinical features and presentations of the 68 patients with JIDH who met our inclusion criteria are listed in Table 1; the average age was 72.41 years (range 48–94 years) and 33 were male (48.53%). The mean interval from initial symptom onset to diagnosis was 31.06 days (range, 0–1089 days; median 5 days). Clinical presentations included tarry stool (83.82%), bloody stool (47.06%), shock (27.94%) and coffee ground vomitus (2.94%). The mean haemoglobin level at presentation was 7.70 g/L. The mean hospital stay was 14.97 days (range, 3–84 days; median, 10.5 days), with patients receiving a mean red blood cell transfusion volume of 11.27 units. The mean follow-up duration was 1589.56 days.

CT scan was the most commonly utilised diagnostic tool for JIDH (N = 60; 88.24%), followed by enteroscopy (N = 39; 57.35%), angiography (N = 14; 20.59%), small bowel barium follow-through (N = 9; 13.24%) and surgery (N = 2; 2.94%). The corresponding diagnostic yields for JIDH were 35.00% (21/60), 87.18% (34/39), 14.29% (2/14), 88.89% (8/9) and 100.00% (2/2), respectively. Rebleeding events were noted in 12 cases (17.65%). All-cause mortality was reported for 8 patients (11.76%), all of whom were older than 71 years; there was only one bleeding-related death (1.47%), with sepsis, malignancy, acute myocardial infarction, respiratory failure and cerebrovascular accident being the other causes.

**Table 1. Characteristics and clinical presentations of participants.**

| Clinical Variables | Data |
|---|---|
| Sex (male/female) | 33/35 |
| Age (years, mean ± SD) | 72.41 ± 8.64 |
| Haemoglobin (g/L, mean ± SD) | 7.70 ± 1.87 |
| Length of Hospital Stay (days, mean ± SD) | 14.97 ± 13.18 |
| RBC Transfusion (units, mean ± SD) | 11.27 ± 10.98 |
| Time to Diagnosis (days, mean ± SD) | 31.06 ± 138.48 |
| Follow-up Duration (days, mean ± SD) | 1589.56 ± 1547.73 |
| Symptoms and Signs, n (%) | |
| Tarry Stool | 57 (83.82%) |
| Bloody Stool | 32 (47.06%) |
| Shock | 19 (27.94%) |
| Coffee Ground Vomitus | 2 (2.94%) |
| Utilised Diagnostic Procedures, n (%) | |
| CT Scan | 60 (88.24%) |
| Enteroscopy | 39 (57.35%) |
| Angiography | 14 (20.59%) |
| Small bowel follow-through | 9 (13.24%) |
| Diagnostic Surgery | 2 (2.94%) |
| Yields of Diagnostic Procedures, n (%) | |
| CT Scan | 21/60 (35.00%) |
| Enteroscopy | 34/39 (87.18%) |
| Angiography | 2/14 (14.29%) |
| Small bowel follow-through | 8/9 (88.89%) |
| Diagnostic Surgery | 2/2 (100.00%) |
| Rebleeding, n (%) | 12 (17.65%) |
| Bleeding-related Mortality, n (%) | 1 (1.47%) |

Abbreviations: CT, computed tomography; GI, gastrointestinal; RBC, red blood cell; SD, standard deviation.

## Comparison between surgically and non-surgically treated groups

The clinical features of 22 patients initially treated surgically were compared to those of 46 patients initially treated non-surgically (Table 2). In the surgical group, hospital stays were longer (24.68 ± 17.40 vs. 10.33 ± 7.02 days, $p < 0.0001$), red blood cell transfusion volumes were higher (15.09 ± 12.15 vs. 8.96 ± 9.91 units, $p = 0.0302$) and rebleeding rates were lower (0% vs. 26.09%, $p = 0.0088$; Fig 1).

Concerning the signs and symptoms, more patients presented with tarry stools in the non-surgical group than in the surgical group (91.30% vs. 82.14%, $p = 0.0162$), but patients in the surgical group had higher proportions of bloody stools (64.29% vs. 32.61%, $p = 0.0006$) and haemorrhagic shock (32.14% vs. 15.22%, $p = 0.0008$). All-cause mortality rates were 13.33% (N = 6) in the non-surgical group and 8.70% (N = 2) in the surgical group, with similar survival curves ($p = 0.4511$; Fig 2).

When comparing initial endoscopic treatment with initial surgical treatment, there were some notable differences. Surgery was used significantly more often in patients with bloody stools ($p = 0.0013$) and haemorrhagic shock ($p = 0.0304$), with this group also having longer hospital stays ($p = 0.0019$). By contrast, the rebleeding rate was higher in the endoscopic treatment group ($p = 0.0048$).

**Table 2. Clinical features of the surgical and non-surgical groups.**

| Clinical Features | Surgery (N = 22) | Non-Surgical Intervention (N = 46) | *p*-value |
|---|---|---|---|
| Age (years, mean ± SD) | 72.55 ± 8.26 | 72.35 ± 8.74 | 0.9287 |
| Haemoglobin (g/L, mean ± SD) | 7.36 ± 2.04 | 7.86 ± 1.79 | 0.3069 |
| Length of Hospital Stay (days, mean ± SD) | 24.68 ± 17.40 | 10.33 ± 7.02 | <0.0001* |
| RBC Transfusion (units, mean ± SD) | 15.09 ± 12.15 | 8.96 ± 9.91 | 0.0302* |
| Time to Diagnosis (days, mean ± SD) | 9.77 ± 25.55 | 41.24 ± 167.09 | 0.3847 |
| Symptoms and Signs, n (%) | | | |
| Tarry Stool | 15 (82.14%) | 42 (91.30%) | 0.0162* |
| Bloody Stool | 17 (64.29%) | 15 (32.61%) | 0.0006* |
| Shock | 12 (32.14%) | 7 (15.22%) | 0.0008* |
| Coffee Ground Vomitus | 1 (3.57%) | 1 (2.17%) | 0.5909 |
| Rebleeding, n (%) | 0 (0.00%) | 12 (26.09%) | 0.0088* |
| Bleeding-related Mortality, n (%) | 0 (0.00%) | 1 (2.17%) | 0.4892 |

Abbreviations: RBC, red blood cell; SD, standard deviation.

## Choice of diagnostic tools and treatments before 2009 and after 2010

From 3 to 2009, with a total number of 35 patients, enteroscopy, CT, angiography, surgery and small bowel follow-through were performed diagnostically in 22 (34.4%), 29 (45.3%), 8 (12.5%), 2 (3.1%) and 4 (6.3%) cases, respectively. The corresponding numbers from 2010 to 2019 in 33 patients were 18 (30.0%), 31 (51.7%), 6 (10.0%), 0 (0.0%) and 5 (8.3%), respectively (Table 3). The choice of diagnostic tool was comparable between each period (*p* = 0.6149) (Fig 3A). From 2000 to 2009, surgery, radiology, medical therapy and endoscopy were used initially in 13 (37.1%), 0

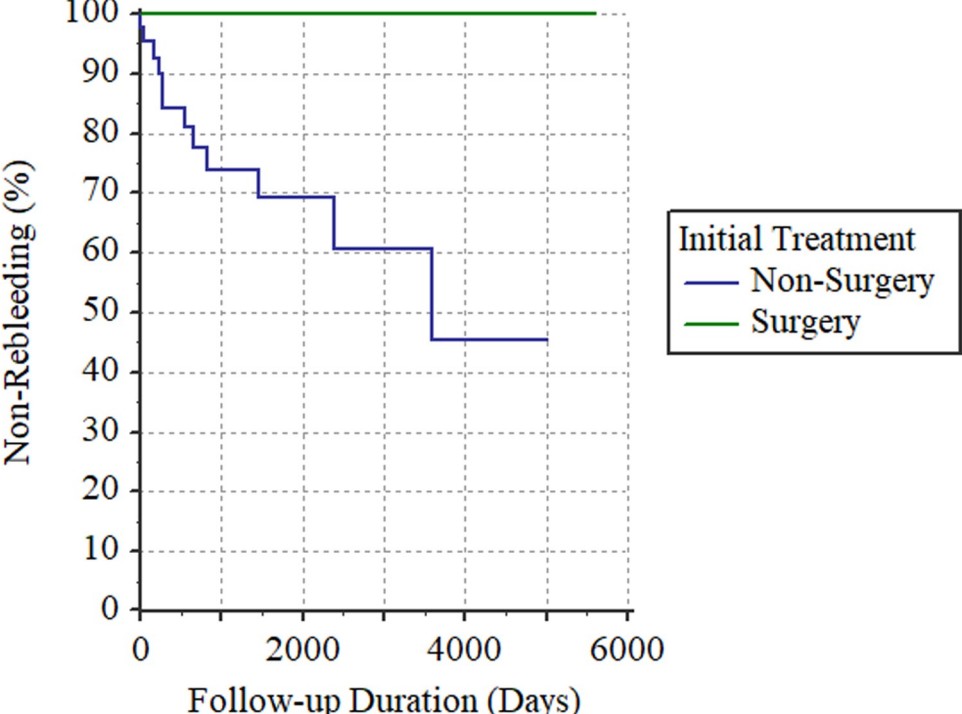

**Fig 1. Comparison of rebleeding rates between patients treated surgically and non-surgically (*p* = 0.0088).**

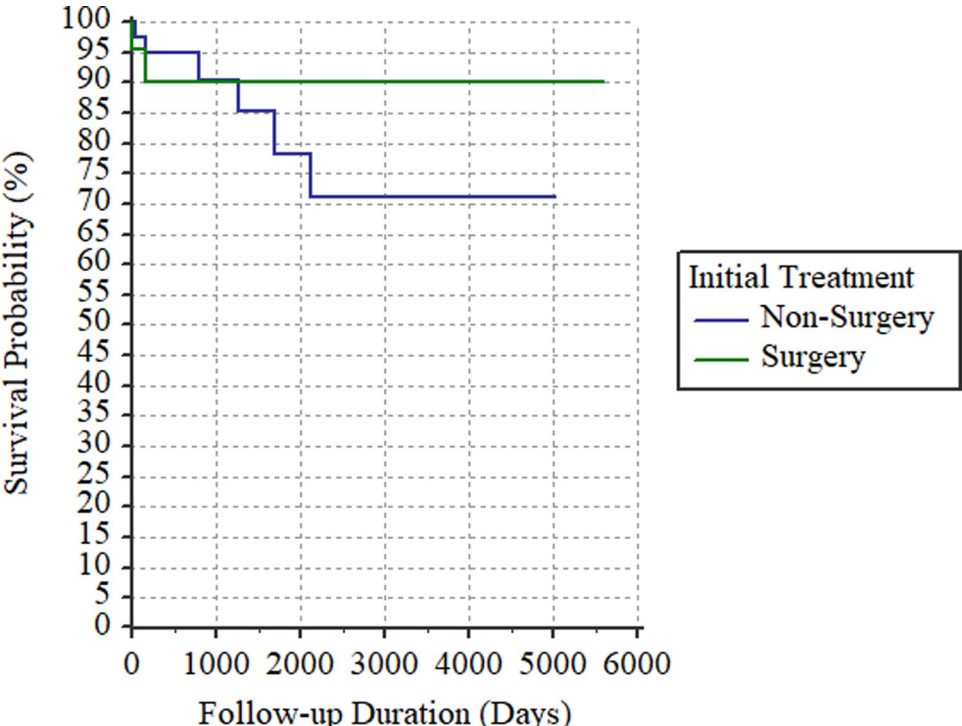

**Fig 2. Comparison of survival curves between patients treated surgically and non-surgically ($p$ = 0.4511).**

(0.0%), 12 (34.3%) and 10 (28.6%) cases, respectively; the corresponding numbers from 2010 to 2019 were 9 (27.3%), 1 (3.0%), 14 (42.4%) and 9 (27.3%), respectively (Table 3). The primary treatment choice did not change significantly between the two periods ($p$ = 0.5984) (Fig 3B).

The outcomes of JIDH after endoscopic and supportive treatments were also studied. The risks of needing subsequent diverticular resection after failed initial endoscopic or supportive treatments were 5.26% and 11.54%, respectively ($p$ = 0.4700). Moreover, the risks of rebleeding ($p$ = 0.5288) and of all-cause mortality ($p$ = 0.6820) were similar in the groups receiving endoscopic and supportive treatments (Table 4).

## Discussion

JIDH is a rare and potentially fatal disease that can be difficult to identify because it is located beyond the reach of regular diagnostic procedures. Previous reports have mainly focused on

**Table 3. Changes in the clinical management of jejunoileal diverticular haemorrhage before 2009 and after 2010.**

| Diagnosis | pre-2009 | post-2010 | Treatments | pre-2009 | post-2010 |
|---|---|---|---|---|---|
| Enteroscopy | 22 (34.4%) | 18 (30.0%) | Surgery | 13 (37.1%) | 9 (27.3%) |
| CT Scan | 29 (45.3%) | 31 (51.7%) | Radiology | 0 (0.0%) | 1 (3.0%) |
| Angiography | 8 (12.5%) | 6 (10.0%) | Medical | 12 (34.3%) | 14 (42.4%) |
| Surgery | 2 (3.1%) | 0 (0.0%) | Endoscopy | 10 (28.6%) | 9 (27.3%) |
| SBFT | 4 (6.3%) | 5 (8.3%) | | | |
| *p*-value | 0.6149 | | *p*-value | 0.5984 | |

Abbreviations: CT, computed tomography; SBFT, Small bowel follow-through.

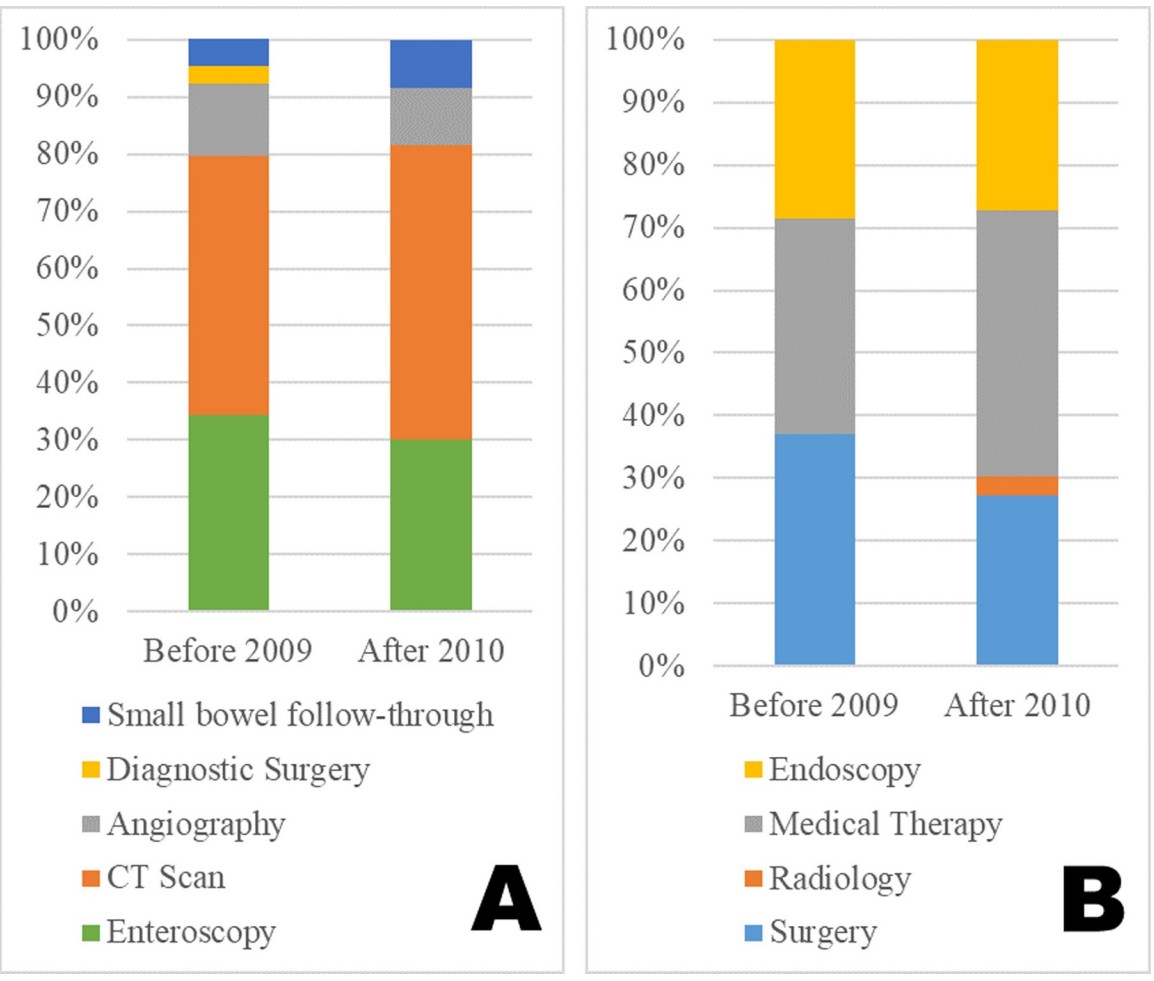

**Fig 3. Clinical management of jejunoileal diverticular haemorrhage before 2009 and after 2010.** (A) Choice of diagnostic tool for jejunoileal diverticular haemorrhage before 2009 and after 2010. (B) Choice of treatment for jejunoileal diverticular haemorrhage before 2009 and after 2010.

radiological diagnosis and surgical management, and although developments in endoscopic techniques should probably have been associated with a change in management, there are no clear data on whether this has occurred. We therefore aimed to report our experience with the

**Table 4. The risks of subsequent surgery, rebleeding event and all-cause mortality after initial endoscopic or supportive treatments.**

|  | Endoscopy (N = 19) | Supportive therapy (N = 26) | *p*-value |
|---|---|---|---|
| Subsequent surgery | 1 | 3 | 0.4700 |
| No subsequent surgery | 18 | 23 | |
| Subsequent surgery rate | 5.26% | 11.54% | |
| Rebleeding | 6 | 6 | 0.5288 |
| Non-rebleeding | 13 | 20 | |
| Rebleeding rate | 31.58% | 30.00% | |
| Mortality (all-cause) | 3 | 3 | 0.6820 |
| Alive | 16 | 23 | |
| Mortality rate | 15.79% | 11.54% | |

management and follow-up of JIDH over recent decades. To the best of our knowledge, this is the largest case series of JIDH in the 21<sup>st</sup> century. Our major findings confirm several clinically relevant features of the disease, whilst also showing that there has been little change in either the diagnostic or the treatment approach between 2000 and 2019.

Clinically, although there was no sex predominance, most patients were elderly. Patients also tended to have low haemoglobin levels and required blood transfusions (N = 65; 95.6%), consistent with the haemorrhagic pathology. Despite the known difficulties in diagnosis, the time from symptom onset to diagnosis was relatively short (median, 5 days). The most common symptoms and signs were tarry stools (83.82%), but bloody stools and shock were present in 47.06% and 27.94%, respectively. These presentations may have raised the clinical suspicion of JIDH.

CT was the most commonly used diagnostic tool for JIDH, although two patients received emergency surgery for massive bleeding, with the diagnosis of JIDH made postoperatively. Surgery yielded 100% diagnostic accuracy, while enteroscopy (87.18%) and small bowel flow-through (88.89%) showed similar performances. Although rebleeding events occurred in 17.65% of patients, bleeding-related mortality was very rare compared with 8.33%–20.59% reported in previous studies, with only one case in our study [10, 16]. We therefore have reservations that the disease is necessarily associated with high mortality, particularly due to delayed diagnosis [17].

Surgical interventions were associated with longer hospital stays and greater blood transfusion volumes, but with lower rebleeding rates. Although hospital stays and blood transfusion reflect disease severity, they also resulted from the surgical intervention. Of note, the rebleeding rate was zero after surgical resection, which should be expected because the lesion is removed; by contrast, non-surgical treatments only seek to achieve haemostasis (Fig 1). Furthermore, based on its multifactorial and insidious pathogenesis, a diverticulum develops chronically [18], and most of the patients present JIDH at an old age. Therefore, in elderly patients, it is less likely to have another bleeding event of JIDH. Tarry stools were mostly presented in the non-surgical group, while bloody stools and shock were mainly present in the surgical group, which are broadly consistent with the management approaches. In Fig 2, the survival curve of surgically treated patients seemed to be above that of non-surgically treated patients; however, the two curves were not statistically difference, indicating comparable all-cause mortality.

When comparing the two decades at our institution, we noted that there had been little change in the diagnostic tools and treatments that were used (Table 3 and Fig 3). We had introduced deep enteroscopy early, in 2004, for the treatment of small intestinal haemorrhage at our institution [19–21]. This had allowed patients to be treated beyond the choice of surgery, dependent on each situation, which may explain why treatment has not changed at our institution.

Although there were no statistical differences between endoscopic and supportive approaches in the rates of subsequent surgery, rebleeding or all-cause mortality, we found a relatively low rate of subsequent surgery in patients treated endoscopically (5.26%) than in patients treated with supportive care (11.54%). We therefore suggest that endoscopic treatments are probably superior to supportive therapy because they achieve haemostasis at higher rates. It is possible that these results would become statistically significant with a larger sample.

Previous studies have indicated that jejunal diverticular haemorrhage has a high mortality rate after conservative treatment (80%) compared with surgical treatment (14%) [22]. However, this based on a summary published in 1969 when endoscopic technology was still in its infancy [16]. Little is known about the changes in management and natural course of this rare disease since the full range of enteroscopy has been introduced. Clinicians today have options other

than surgery or supportive therapy, and our results provides information about experiences with all treatment options in a long-term and relatively large sample of patients with JIDH.

Currently, the gold standard of diagnosis and treatment are small bowel flow-through radiography followed by limited surgical resection of the involved intestine [8]. However, small bowel flow-through was only performed in a minority of patients with JIDH, and it was less useful than either CT or enteroscopy. Although there was no rebleeding associated with surgery, which was reserved for more severe cases, this approach was highly invasive and associated with significant healthcare expenditure. This included longer hospitalisations (double) and larger blood transfusion volumes.

Despite the risk of rebleeding, mortality after initial non-surgical treatment was very rare, suggesting that long-term outcomes could be acceptable in JIDH. Consistent with our previous report [4], approximately two-thirds of patients were treated non-surgically, which may be due to improvements in diagnosis and treatment over recent decades. Although we still advocate surgery as the standard treatment, endoscopy clearly has a role as an alternative that may be appropriate in certain clinical situations. Indeed, surgical intervention should be preferred for the relatively young or those with low haemoglobin levels, bloody stools and haemorrhagic shock; by contrast, non-surgical interventions may be favoured for older patients and those with less severe disease or contraindications to surgery. Of course, patients receiving non-surgical treatments will need to be informed of the possibility of subsequent surgical intervention if non-surgical treatments fail.

There are some limitations in the present study. Most notably, this was a retrospective analysis, and the clinical evaluations, diagnoses and treatments were operator dependent and could not be standardised. Although the case number was adequate for gaining a better understanding of JIDH, it was insufficient to allow statistical power for subgroup analysis (e.g. comparison of outcomes between endoscopic and supportive therapies).

In conclusion, we have described our experiences with the management of JIDH over the last two decades, during which we found no significant changes in diagnostic or treatment approaches. In patients with gastrointestinal bleeding, after excluding upper and lower gastrointestinal haemorrhage, enteroscopy can be very helpful. We recommend that surgery should remain the treatment of choice for JIDH, but our experience also indicates that endoscopic treatment could increasingly be considered as an alternative option in certain patient groups.

## Supporting information

**S1 Data.**
(PDF)

## Author Contributions

**Conceptualization:** Hsu-Heng Yen.

**Data curation:** Hsuan-An Su, Yu-Chun Hsu, Hsu-Heng Yen.

**Formal analysis:** Hsuan-An Su, Yu-Chun Hsu, Fu-Yuan Siao, Hsu-Heng Yen.

**Funding acquisition:** Hsu-Heng Yen.

**Investigation:** Hsuan-An Su, Yu-Chun Hsu, Fu-Yuan Siao, Hsu-Heng Yen.

**Supervision:** Hsu-Heng Yen.

**Writing – original draft:** Hsuan-An Su, Yu-Chun Hsu.

**Writing – review & editing:** Fu-Yuan Siao, Hsu-Heng Yen.

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
