## [Decision Letter · Decision Letter 0]

27 Apr 2020

PONE-D-20-07441

Diagnosis and Management of Jejunal–ileal Diverticular Haemorrhage: An Update on the Experience in a Single Centre

PLOS ONE

Dear Dr. Yen,

Thank you for submitting your manuscript to PLOS ONE. After careful consideration, we feel that it has merit but does not fully meet PLOS ONE’s publication criteria as it currently stands. Therefore, we invite you to submit a revised version of the manuscript that addresses the points raised during the review process.

We would appreciate receiving your revised manuscript by Jun 11 2020 11:59PM. To enhance the reproducibility of your results, we recommend that if applicable you deposit your laboratory protocols in protocols.io, where a protocol can be assigned its own identifier (DOI) such that it can be cited independently in the future. For instructions see: http://journals.plos.org/plosone/s/submission-guidelines#loc-laboratory-protocols

We look forward to receiving your revised manuscript.

Kind regards,

Chun Chieh Yeh, M.D., Ph.D.

Academic Editor

PLOS ONE

Additional Editor Comments:

1.Please revise the formate of the article based on basic formula of scientific article

2.please response the reviewers' comments specifically and appropriately. We would give final decision after your timely revisions.

Reviewers' comments:

Reviewer's Responses to Questions

**Comments to the Author**

1. Is the manuscript technically sound, and do the data support the conclusions?

Reviewer #1: Yes

Reviewer #2: Partly

2. Has the statistical analysis been performed appropriately and rigorously? 

Reviewer #1: Yes

Reviewer #2: Yes

3. Have the authors made all data underlying the findings in their manuscript fully available?

Reviewer #1: Yes

Reviewer #2: No

4. Is the manuscript presented in an intelligible fashion and written in standard English?

Reviewer #1: Yes

Reviewer #2: Yes

5. Review Comments to the Author

Reviewer #1: To the authors,

This manuscript is a very interesting and valuable study in the GI field. As we know, the jejunoileal diverticular bleeding is not common in the clinical practice, especially in the Western countries. By the contrast, the jejunoileal diverticular bleeding is more common in the Asian countries compared to the Western countries.

I have some questions for your manuscript

Q1. In the section: Enteroscopy is the primary diagnostic approach for jejunoileal diverticular haemorrhage (JIDH) and includes endoscopy of the small intestines [6], small bowel follow-through studies, computed tomography (CT), angiography and technetium red cell-tagged scans that are beyond the reach of esophagogastroduodenoscopy and colonoscopy [1]. Its meaning is not very clear to readers. Can you rewrite them ?

Q2. Your manuscript mentioned "Investigation revealed coexisting gastrointestinal diverticula in the duodenum (23.53%) and colon (14.71%)". Can I ask the diagnostic tool for your coexisting gastrointestinal diverticula?

Q3. In your manuscript, you said that deep enteroscopy was introduced at your institution in 2004 for the treatment of small intestinal haemorrhage. What is your reason for the decreased rate in diagnostic enteroscopy after 2010? Moreover, what is your diagnostic enteroscopy before 2004?

Q4. In the diagnostic modalities of jejunoileal diverticula, your described the total number was 64 before 2009, and 60 after 2010. Does the number mean the patients numbers? (your total enrolled patients were 68)

Thanks ,

Reviewer #2: This is a retrospective study about jejunoileal diveticular hemorhage. the study included small number of patients- 68 only.

some major revisions need to be done:

1- the abstract should be fragmented into introduction, methods, results and conclusion- in breif as the whole article.

2- in the introdcution section: enteroscopy is an endoscopy of the small bowel and does not include small bowel follow through, CT scan, angiography and Tc RBC scan- these are a different radiological dignostic tools, and not as written that are part of the enteroscopy.

3- following the introduction, methods and materials should be written before the results.

4- the most utilised diagnostic procedure was CT scan according to the authors- the authors means CT scan or CT angiography scan? and wether CT angiography was used as a diagnostic procedure.

5- what were the findings on the different diagnostic tools?

6- 22 patiens were treated initially by surgical means- but afterward, surgery was used in total of 7 patients for diagnosis... this means that 15 patients actually underwent other dianostic procedures- what are these procedures? and what was the indication for operation?

7- table 3 should include the total number of patients before 2009 and total number after.

6. PLOS authors have the option to publish the peer review history of their article (what does this mean?). If published, this will include your full peer review and any attached files.

Reviewer #1: No

Reviewer #2: No

---

## [Author Response · Author response to Decision Letter 0]

1 May 2020

Chun Chieh Yeh, M.D., Ph.D.

Academic Editor

PLOS ONE

Dear Editor,

We appreciate your editorial comments, as well as those of the reviewers, concerning our manuscript. Based on these comments, we have made several revisions to our manuscript, which is resubmitted for your consideration. The manuscript has also been revised according to the PLOS ONE's style requirements. If there is anything needing to be further improved, please do not hesitate to inform us at your earliest convenience. Your assistance is highly appreciated. We look forward to your message.

The followings are point-by-point responses to the comments.

Response: The manuscript has been revised per the journal’s requirement. If there is anything further required, please do not hesitate to inform us. We are very happy to cooperate.

Response: We understand the policy of the journal. All authors agreed to provide de-identified raw data as supporting information from which all of the results were derived. The phrase “data not shown” has been removed.

Reviewers' comments:

Reviewer's Responses to Questions

Comments to the Author

1. Is the manuscript technically sound, and do the data support the conclusions?

Reviewer #1: Yes

Reviewer #2: Partly

2. Has the statistical analysis been performed appropriately and rigorously?

Reviewer #1: Yes

Reviewer #2: Yes

3. Have the authors made all data underlying the findings in their manuscript fully available?

Reviewer #1: Yes

Reviewer #2: No

Response: All authors agreed to provide de-identified raw data as supporting information from which all of the results were derived.

4. Is the manuscript presented in an intelligible fashion and written in standard English?

Reviewer #1: Yes

Reviewer #2: Yes

5. Review Comments to the Author

Reviewer #1: 

To the authors,

This manuscript is a very interesting and valuable study in the GI field. As we know, the jejunoileal diverticular bleeding is not common in the clinical practice, especially in the Western countries. By the contrast, the jejunoileal diverticular bleeding is more common in the Asian countries compared to the Western countries. 

I have some questions for your manuscript

Q1. In the section: Enteroscopy is the primary diagnostic approach for jejunoileal diverticular haemorrhage (JIDH) and includes endoscopy of the small intestines [6], small bowel follow-through studies, computed tomography (CT), angiography and technetium red cell-tagged scans that are beyond the reach of esophagogastroduodenoscopy and colonoscopy [1]. Its meaning is not very clear to readers. Can you rewrite them ?

Response: All authors thank the reviewer’s correction. We apologize for the misleading and somewhat mistaken sentence. The sentence has been rewritten as follows, “Enteroscopy is the current primary endoscopic diagnostic and therapeutic approach for obscure gastrointestinal bleeding [6] including jejunoileal diverticular haemorrhage (JIDH). Alternative non-endoscopic diagnostic tools, including small bowel follow-through studies, computed tomography (CT), angiography and technetium red cell-tagged scans, are used when enteroscopy is not available [1],” as in line 49-53.

Q2. Your manuscript mentioned "Investigation revealed coexisting gastrointestinal diverticula in the duodenum (23.53%) and colon (14.71%)". Can I ask the diagnostic tool for your coexisting gastrointestinal diverticula?

Response: We thank the reviewer for the question. Because of higher incidence as well as anatomical accessibility of the gastro-duodenal and colonic hemorrhage, patients with gastrointestinal bleeding will received either esophagogastroduodenoscopy or colonoscopy before further investigation for small intestinal bleeder. Hence, those duodenal and colonic diverticula were found by esophagogastroduodenoscopy or colonoscopy. Please find line 116 where we have added a sentence to address this concern.

Q3. In your manuscript, you said that deep enteroscopy was introduced at your institution in 2004 for the treatment of small intestinal haemorrhage. What is your reason for the decreased rate in diagnostic enteroscopy after 2010? Moreover, what is your diagnostic enteroscopy before 2004?

Response: The authors thank the reviewer for the question. The management of obscure GI bleeding changed over the past two decades. For example, in the AGA medical position statement (Gastroenterology. 2000 Jan;118(1):197-201. DOI: 10.1016/s0016-5085(00)70429-x), there was no role of CT scan in diagnosis/management for OGIB.

The situation changed after the introduction of deep enteroscopy. In fact, deep enteroscopy was introduced to Taiwan and our institution in 2004. Before 2004, we performed push endoscopy which could only visualize the proximal jejunum at most. As the resolution and feasibility of computed tomography angiography (CTA) improved (Yen HH et al. World J Gastroenterol. 2012 Feb 21;18(7):692-7. doi: 10.3748/wjg.v18.i7.692), patients usually received non-invasive diagnostic procedure such as CTA first, followed by invasive diagnostic procedures such as enteroscopy.

Q4. In the diagnostic modalities of jejunoileal diverticula, your described the total number was 64 before 2009, and 60 after 2010. Does the number mean the patients numbers? (your total enrolled patients were 68)

Response: All authors thank the reviewer for the correction. The number “64” and “60” were the sums of the counts of the diagnostic procedures. They were not referred to as the number of patients, and they were therefore negligible. We would like to remove the row of the table in order not to cause confusion as well as unnecessity. The total numbers of patients before 2009 and after 2010 were 35 and 33, respectively, which we would like to add in line 153 and 156.

Reviewer #2:

This is a retrospective study about jejunoileal diveticular hemorhage. the study included small number of patients- 68 only.

some major revisions need to be done:

1- the abstract should be fragmented into introduction, methods, results and conclusion- in breif as the whole article.

Response: The authors thank the reviewer for the correction. The abstract has been rewritten as a structured abstract.

2- in the introdcution section: enteroscopy is an endoscopy of the small bowel and does not include small bowel follow through, CT scan, angiography and Tc RBC scan- these are a different radiological dignostic tools, and not as written that are part of the enteroscopy.

Response: All authors thank the reviewer’s correction. We apologize for the misleading and somewhat mistaken sentence. The sentence has been rewritten as follows, “Enteroscopy is the current primary endoscopic diagnostic and therapeutic approach for obscure gastrointestinal bleeding [6] including jejunoileal diverticular haemorrhage (JIDH). Alternative non-endoscopic diagnostic tools, including small bowel follow-through studies, computed tomography (CT), angiography and technetium red cell-tagged scans, are used when enteroscopy is not available [1],” as in line 49-53.

3- following the introduction, methods and materials should be written before the results.

Response: The authors thank the reviewer for the correction. The “materials and methods” section has been shifted right after the “introduction” section.

4- the most utilised diagnostic procedure was CT scan according to the authors- the authors means CT scan or CT angiography scan? and wether CT angiography was used as a diagnostic procedure.

Response: All authors thank the reviewer for the question. In our institution, abdominal CT scan indicated for GI bleeding is conducted with CT angiography protocol (Yen HH et al. World J Gastroenterol. 2012 Feb 21;18(7):692-7. DOI: 10.3748/wjg.v18.i7.692), because non-contrast CT scan is not helpful. Therefore, in the present study, all CT scans were performed as CT angiography. We did not clarify this point in the manuscript and we believe the same question would be raised by readers. We would like to add a sentence, “All of the CT scans in the present study were performed with CT angiography protocol”, in line 89 and 90, to avoid such concern.

5- what were the findings on the different diagnostic tools?

Response: Different diagnostic tools in the present study included enteroscopy, CT scan (CT angiography), angiography, surgery, and small bowel flow-through study. By enteroscopy, bleeders or lesions could be directly visualized with in small intestines. CTA scan is able to diagnose and localize the nature of the bleeder, such as tumor or diverticulum. Angiography is used to localize the bleeder with contrast extravasation, yet with inferior diagnostic capability to outline non-vascular lesions or structures. Exploratory surgery was commonly conducted under emergency and could directly expose the lesions followed by surgical repair. Finally, small bowel flow-through study utilizes contrast medium to delineate the intestinal lumen, and therefore could facilitate diagnosis of bowel structural abnormality such as diverticulum, unable to confirm the presence of bleeding or the nature of the bleeder.

6- 22 patiens were treated initially by surgical means- but afterward, surgery was used in total of 7 patients for diagnosis... this means that 15 patients actually underwent other dianostic procedures- what are these procedures? and what was the indication for operation?

Response: The authors thank the reviewer’s comment. Overall, 22 patients were treated with surgery, and with all due respect, diagnostic surgery had been performed in 2, but not 7, patients. In the other 20 patients, these diagnostic procedures included CT scan in 13 patients, angiography in 5 patients, and endoscopy in 2 patients. The indications for surgery were jejunoileal diverticular bleeding based on individualized clinical situations.

7- table 3 should include the total number of patients before 2009 and total number after.

Response: The authors thank the reviewer’s suggestion. The total numbers of patients before 2009 and after 2010 were 35 and 33, respectively. However, a patient could have undergone more than one diagnostic procedures. We are afraid that if the total number of patient be written in Table 3, the readers might be confused with the total number of the patients or the procedures. Therefore, please allow us to remove the total number from Table 3. Alternatively, we would like to add the total number of patients in the text, as in line 153 and 156.

Thank you for the opportunity to resubmit this manuscript for consideration of publication in PLOS ONE. If you have any questions or comments regarding this manuscript, please do not hesitate to contact us by mail at our correspondence address, by fax at +886-4-7228289, by telephone at +886-4-7238595ext5501, or by e-mail at 91646@cch.org.tw

Sincerely, 

Hsu-Heng Yen M.D 

Department of Gastroenterology 

Changhua Christian Hospital, Taiwan.

---

## [Decision Letter · Decision Letter 1]

14 May 2020

PONE-D-20-07441R1

Diagnosis and management of jejunoileal diverticular haemorrhage: an update on the experience in a single centre

PLOS ONE

Dear Dr. Yen,

Thank you for submitting your manuscript to PLOS ONE. After careful consideration, we feel that it has merit but does not fully meet PLOS ONE’s publication criteria as it currently stands. Therefore, we invite you to submit a revised version of the manuscript that addresses the points raised during the review process.

We would appreciate receiving your revised manuscript by Jun 28 2020 11:59PM. To enhance the reproducibility of your results, we recommend that if applicable you deposit your laboratory protocols in protocols.io, where a protocol can be assigned its own identifier (DOI) such that it can be cited independently in the future. For instructions see: http://journals.plos.org/plosone/s/submission-guidelines#loc-laboratory-protocols

We look forward to receiving your revised manuscript.

Kind regards,

Chun Chieh Yeh, M.D., Ph.D.

Academic Editor

PLOS ONE

Additional Editor Comments (if provided):

Please response our reviewer's concerns and make corresponding revisions again.

Reviewers' comments:

Reviewer's Responses to Questions

**Comments to the Author**

1. If the authors have adequately addressed your comments raised in a previous round of review and you feel that this manuscript is now acceptable for publication, you may indicate that here to bypass the “Comments to the Author” section, enter your conflict of interest statement in the “Confidential to Editor” section, and submit your "Accept" recommendation.

Reviewer #1: All comments have been addressed

Reviewer #2: All comments have been addressed

2. Is the manuscript technically sound, and do the data support the conclusions?

Reviewer #1: Yes

Reviewer #2: Yes

3. Has the statistical analysis been performed appropriately and rigorously? 

Reviewer #1: Yes

Reviewer #2: Yes

4. Have the authors made all data underlying the findings in their manuscript fully available?

Reviewer #1: Yes

Reviewer #2: Yes

5. Is the manuscript presented in an intelligible fashion and written in standard English?

Reviewer #1: Yes

Reviewer #2: Yes

6. Review Comments to the Author

Reviewer #1: Dear Authors,

Thanks for your good responses for our reviewer's comments

I still have some questions for your manuscript.

Comment 1: The ratio of coexisting duodenal diverticulum or colonic diverticulum is for all patients number. Do your all patients undergo the colonoscopy and EGD before the diagnosis of jejunoileal diverticular bleeding? Because some patients may experience emergency surgery after CT-angiography, small bowel series or enteroscopy (including push endoscopy) without colonoscopy or EGD. Do your raw data have complete EGD and colonoscopy for every patient?

I think the coexisting duodenal or colonic diverticulum is not necessary for your manuscript

Comment 2: You mentioned that your hospital had push enteroscopy and new enteroscopy to diagnose the jejunoileal diverticular bleeding. I suggest your should describe it clearly in your diagnostic modality and your tables.

Best regard

Reviewer #2: following the major revisions made, I have no comments to the authors.

authors did change the structure of the manuscript in a way that is suitable for retroscpective study. moreover, they introduce the data and the statistics as requested.

they also answer all the questions that I already asked.

7. PLOS authors have the option to publish the peer review history of their article (what does this mean?). If published, this will include your full peer review and any attached files.

Reviewer #1: Yes: Jen-Wei Chou

Reviewer #2: No

---

## [Author Response · Author response to Decision Letter 1]

15 May 2020

Chun Chieh Yeh, M.D., Ph.D.

Academic Editor

PLOS ONE

05/15/2020

Dear Editor,

We appreciate your editorial comments, as well as those of the reviewer Dr. Jen-Wei Chou and reviewer #2, concerning our revised manuscript. Based on these comments, we have made further revisions to our manuscript, which is resubmitted for your consideration. Your assistance is highly appreciated. We look forward to your message.

The followings are point-by-point responses to the comments:

Reviewer #1: 

Dear Authors,

Thanks for your good responses for our reviewer's comments

I still have some questions for your manuscript.

Comment 1: The ratio of coexisting duodenal diverticulum or colonic diverticulum is for all patients number. Do your all patients undergo the colonoscopy and EGD before the diagnosis of jejunoileal diverticular bleeding? Because some patients may experience emergency surgery after CT-angiography, small bowel series or enteroscopy (including push endoscopy) without colonoscopy or EGD. Do your raw data have complete EGD and colonoscopy for every patient?

I think the coexisting duodenal or colonic diverticulum is not necessary for your manuscript.

Response: All authors appreciate the reviewer for the correction very much. According to our raw data, not all of the subjects had undergone colonoscopy or EGD. Therefore, the ratio of coexisting diverticulum we presented based on all patient number was inappropriate. All authors agree with the reviewer’s opinion that such data is not necessary for our study, and any content relevant to the “coexisting duodenal or colonic diverticulum” has been removed, as in LINE 87, 117, 191, and in Table 1.

Comment 2: You mentioned that your hospital had push enteroscopy and new enteroscopy to diagnose the jejunoileal diverticular bleeding. I suggest your should describe it clearly in your diagnostic modality and your tables.

Response: All authors thank the reviewer for the suggestion. We think that it is better to be added in the “Materials and Methods” section, as in LINE 90-93: “The methods of enteroscopy include push endoscope (SIF-Q140, Olympus Co., Japan) performed in 5 cases before 2004, and double-balloon endoscope (EN-450P5 or EN-450T5, Fujinon Co., Japan) performed in the rest of the cases after 2004 in our institution.”

Reviewer #2: following the major revisions made, I have no comments to the authors.

authors did change the structure of the manuscript in a way that is suitable for retroscpective study. moreover, they introduce the data and the statistics as requested.

they also answer all the questions that I already asked.

Response: All authors thank the reviewer for reviewing our revised manuscript and the positive feedback. Your assistance is highly appreciated.

Thank you for the opportunity to resubmit this manuscript for consideration of publication in PLOS ONE. If you have any questions or comments regarding this manuscript, please do not hesitate to contact us by mail at our correspondence address, by fax at +886-4-7228289, by telephone at +886-4-7238595ext5501, or by e-mail at 91646@cch.org.tw

Sincerely, 

Hsu-Heng Yen M.D 

Department of Gastroenterology 

Changhua Christian Hospital, Taiwan.

---

## [Decision Letter · Decision Letter 2]

27 May 2020

Diagnosis and management of jejunoileal diverticular haemorrhage: an update on the experience in a single centre

PONE-D-20-07441R2

Dear Dr. Yen,

We are pleased to inform you that your manuscript has been judged scientifically suitable for publication and will be formally accepted for publication once it complies with all outstanding technical requirements.

With kind regards,

Chun Chieh Yeh, M.D., Ph.D.

Academic Editor

PLOS ONE

Additional Editor Comments (optional):

Thanks for your prompt and appropriate response to the comments raised by our invited reviewers. We consider the manuscript could be accepted at its current content.

Reviewers' comments:

Reviewer's Responses to Questions

**Comments to the Author**

1. If the authors have adequately addressed your comments raised in a previous round of review and you feel that this manuscript is now acceptable for publication, you may indicate that here to bypass the “Comments to the Author” section, enter your conflict of interest statement in the “Confidential to Editor” section, and submit your "Accept" recommendation.

Reviewer #1: All comments have been addressed

2. Is the manuscript technically sound, and do the data support the conclusions?

Reviewer #1: Yes

3. Has the statistical analysis been performed appropriately and rigorously? 

Reviewer #1: Yes

4. Have the authors made all data underlying the findings in their manuscript fully available?

Reviewer #1: Yes

5. Is the manuscript presented in an intelligible fashion and written in standard English?

Reviewer #1: Yes

6. Review Comments to the Author

Reviewer #1: Dear authors,

Thanks for your revisions by my comments. Now, I have no any comments for your manuscript

7. PLOS authors have the option to publish the peer review history of their article (what does this mean?). If published, this will include your full peer review and any attached files.

Reviewer #1: No

---

## [Editor Report · Acceptance letter]

10 Jun 2020

PONE-D-20-07441R2 

Diagnosis and management of jejunoileal diverticular haemorrhage: an update on the experience in a single centre 

Dear Dr. Yen:

I'm pleased to inform you that your manuscript has been deemed suitable for publication in PLOS ONE. Congratulations! Your manuscript is now with our production department. 

Kind regards, 

on behalf of

Dr. Chun Chieh Yeh 

Academic Editor

PLOS ONE